# Osteoradionecrosis of the Temporal Bone as a Rare Cause of Facial Nerve Palsy

**DOI:** 10.3390/diagnostics12051021

**Published:** 2022-04-19

**Authors:** Florian Schmidt, Katy Bradley, Gerd Fabian Volk

**Affiliations:** 1ENT Department, Portsmouth Hospitals University, Portsmouth PO6 3LY, UK; 2Klinik für HNO-Heilkunde, Evangelisches Krankenhaus Düsseldorf, 40217 Düsseldorf, Germany; 3Oncology Department, Portsmouth Hospitals University, Portsmouth PO6 3LY, UK; katy.bradley2@nhs.net; 4Oncology Department, Brighton and Sussex University Hospitals, Brighton BN2 5BE, UK; 5Department of Otorhinolaryngology, Jena University Hospital, Am Klinikum 1, 07747 Jena, Germany; fabian.volk@med.uni-jena.de; 6Facial-Nerve-Center Jena, Jena University Hospital, Am Klinikum 1, 07747 Jena, Germany; 7Center of Rare Diseases Jena, Jena University Hospital, Am Klinikum 1, 07747 Jena, Germany

**Keywords:** facial nerve disorders, osteoradionecrosis, temporal bone, radiotherapy, head and neck cancer, mastoidectomy, surgery, facial palsy

## Abstract

We present a case of a 69-year-old male who presented with acute left facial nerve palsy, serous bloody otorrhea, otalgia, and exposed necrotic bone on the floor of his left ear canal. His medical history revealed a left canal wall-down (CWD) mastoidectomy thirty years ago. Subsequently, twenty years later, he received primary chemoradiotherapy for tonsil cancer on the same side. The patient’s medical history, the typical clinical picture, and a comprehensive diagnostic workup, including imaging modalities and electrophysiology, finally led to a diagnosis of osteoradionecrosis of the temporal bone (ORNTB), with secondary facial nerve palsy. The facial nerve, unfortunately, did not recover and treatment remained conservative, as per the patient’s preference. ORNTB is a rare, delayed complication after radiotherapy for head and neck cancer, which occurs after about 8 years and a minimum of 41.8 Gray of radiation to the affected area. Facial nerve palsy in ORNTB is rare, with only 2.9% of patients experiencing it, but, in our particular case, the patient had undergone an additional CWD mastoidectomy. The treatment options need to be personalized and aimed at symptom control. There should be awareness of the condition among ENT specialists, especially during head and neck cancer follow-ups, and in patients who have had mastoidectomy and radiotherapy affecting the ipsilateral temporal bone.

## 1. Introduction

Acute facial nerve palsy is, most commonly, an idiopathic condition (Bell’s palsy). Other causes of facial nerve palsies include a wide range of disease processes, especially of the parotid gland and temporal bone. Thorough analysis of the patient’s medical history, examination, and diagnostics are required to identify and treat the underlying condition. The following case reports on a rare condition, osteoradionecrosis of the temporal bone. This occurs several years post radiotherapy. It can affect the facial nerve, leading to facial nerve palsy. This case raises awareness of this rare condition among clinicians involved in facial nerve disorders, otology, and head and neck cancer treatment.

## 2. Case Description

A 69-year-old male was referred to our emergency ENT service with a three-week history of left facial nerve palsy. He also reported serous bloody otorrhea and ongoing otalgia. In primary care, he had already received 60 mg of prednisolone over ten days, amoxicillin and topical ciprofloxacin, alongside corneal protection.

Clinical examination revealed left lower motor neuron facial nerve paresis of a House-Brackmann grade V. The left ear showed widely exposed necrotic bone on the floor of the ear canal and over the facial ridge towards the mastoid cavity, with debris (Figure 1).

His medical history revealed a left canal wall-down (CWD) mastoidectomy about thirty years ago. Subsequently, twenty years later (ten years ago), a left-sided squamous cell carcinoma of his tonsil was diagnosed, staged T3N1M0 (7th Edition of the AJCC TNM staging system). He underwent curativeinduction chemotherapy, including fluorouracil, docetaxel, and cisplatin, followed by a radiation dose of 66 Gray (Gy) to his left oropharynx and neck, with concurrent cisplatin (Figure 2). Seven years after his tonsil cancer, he had complained of intermittent otalgia and discomfort in his left periauricular, and was treated with microsuction and topical antibiotics for suspected mastoid cavity infections. Further review of his medical history revealed type II diabetes mellitus, which was medicated orally with diabetic retinopathy and peripheral neuropathy.

At this point, the considered differential diagnosis included complicated acute otitis media, recurrent cholesteatoma, and malignancy of the middle ear and/or external auditory canal. Initially, however, the main diagnosis was thought to be malignant otitis externa (MOE), considering his diabetes.

The following work-up included microbiology from the ear, which revealed no bacterial growth, but some Candida species. A tissue biopsy from the ear canal reported hyperkeratosis, without significant inflammation, fungal organisms, dysplasia, or malignancy. His pure-tone audiogram revealed severe-to-profound pantonal hearing loss on the left (85–110 dB) and presbycusis (10–70 dB) on the right. He then underwent computed tomography (CT) scanning of his petrous bones and neck, reporting non-specific soft tissue in the mastoid cavity, post-surgical changes after previous CWD mastoidectomy, with removal of the ossicular chain, and patchy bony erosion of the external ear canal floor, as well as fatty changes in the left parotid gland, as compared to the right. There were normal appearances of the inner ear structures and the right temporal bone, and there was no evidence of local tumor recurrence or metastatic disease. A technetium uptake scan reported increased osteoblastic activity around the bony portion of the left external auditory canal. Magnetic resonance imaging (MRI) showed enhancement of the left facial nerve and small traces of diffusion in isointense/hyperintense material that was present, which involved the operated left mastoid cavity along the roof, floor, and medial aspect (Figure 3).

He was commenced on a local MOE protocol with oral ciprofloxacin and intravenous ceftazidime for a total of six weeks, and received careful aural toilet and corneal protection. His diabetes was treated with insulin.

After completing treatment, his facial nerve did improve to House–Brackmann grade IV, and the ear was dry and pain free. A needle electromyography only revealed very limited voluntary facial muscle activity in the left orbicularis oculi muscle and no voluntary motor unit activity in any other tested facial muscle on the left side of the face, but also showed no signs of denervation.

Altogether, symptom control was achieved, leaving residual facial nerve palsy and persistent exposed necrotic bone in the ear canal. Based on his medical history, and subsequent investigations, the diagnosis of osteoradionecrosis of the temporal bone (ORNTB), with secondary facial nerve palsy, was established. It must be noted that the correct diagnosis was reached only three years after the development of the first symptoms. At that point, surgery, in the form of subtotal petrosectomy with vascularised flap obliteration, was offered, but, due to comorbidities and the patient’s preference, conservative management was pursued, with regular aural care.

## 3. Discussion

Osteoradionecrosis of the temporal bone after radiotherapy (RT) was first described in 1926 as bony osteitis [1]. It is believed to be “due to radiation damage to blood vessels, causing osteocyte loss, fibrosis, hypocellularity and fatty degeneration of bone” [2]. The tympanic bone seems to be susceptible to radiation effects because of its “superficial position, thin epithelium and resident flora” [3]. A systematic review, including thirty-eight studies, encompassing 364 patients with ORNTB, reported that the mean lag time between radiotherapy and ORN symptoms was 7.9 years (range 6 months to 48 years) [3,4]. The most commonly radiated primary sites leading to the condition were the nasopharynx (36.8%), parotid (20.2%), and external auditory canal (16.3%) [4], unlike our case of an oropharynx primary. The mean radiation dose to the affected temporal bone was reported as 53.1 Gy, with a minimum of 41.8 Gy [3].

Despite differences in the initial treatment approaches for advanced stage III/IV head and neck cancers (especially oropharynx primaries, as described here), such as primary surgery, being more common in Europe and Germany compared to the United States and the United Kingdom, where organ preservation primary chemoradiation protocols are more often employed [5,6], radiation therapy plays a crucial part in the multimodal management of these tumors. The doses of curative primary radiation to gross tumor sites have to exceed 60 Gy (usually a dose equivalent of 70 Gy in 35 fractions), and even elective volumes in the neck range between 30 and 60 Gy. Postoperative radiation to high-risk areas employs volumes between 60 and 66 Gy. Furthermore, in the Anglo-American region, even early-stage oropharyngeal tumors (T1, T2) might be treated with radical radiation as a single modality.

The key clinical finding in ORNTB is visible, exposed necrotic bone in the external auditory canal [3]. Other common symptoms are otorrhea (33.3%), hearing loss (29.1%) and otalgia (17%) [4]. In contrast, facial nerve palsy, as a feature of ORNTB, is rarely reported, with only 2.9% reported cases in a large review [4]. This is likely due to the deep, bony protected course of the facial nerve within the temporal bone. In our case, we think that altered anatomy after the previous CWD mastoidectomy was a risk factor, with bony erosion leading to an exposed facial nerve in the mastoidal segment and subsequent nerve damage.

Facial nerve palsy was the critical symptom in our case, initiating the described comprehensive diagnostic procedure, which was needed to finally reach the correct diagnosis. For acute lower facial nerve palsies, there is no international standard for the optimal extend of diagnostics. However, for ENT specialists, a thorough clinical examination of the ear and parotid gland should be mandatory. Furthermore, the degree of facial nerve impairment should be recorded, which, in clinical routine, is often conducted using subjective measures, such as the House–Brackmann Scale or the Sunnybrook Facial Grading Scale, as objective means, like automated image analysis tools, are still being developed. Electromyography in this context can be helpful in delineating the degree of facial nerve injury and in monitoring the condition. CT scanning of the temporal bone as initial imaging provided, in our case, important information about the regional bony anatomy and extent of disease. Nuclear medicine uptake scans, often used to monitor the treatment response for malignant otitis externa, can provide information about acute inflammation via the accumulation of tracers in the inflamed area. Finally, MRI scanning was rather unspecific for ORNTB, but was helpful in this case to exclude other causes of non-recovering facial nerve palsy.

Different classification systems of ORNTB have been introduced. In 1975, Ramsden et al. distinguished between localized disease, confined to the EAC and tympanic bone, with mild otalgia and otorrhea, and diffuse disease, with extensive necrosis of the temporal bone, with involvement of adjacent structures, such as the labyrinth, facial nerve, TMJ, or brain, with severe otalgia and profuse otorrhea [7]. In 2011, Morrissey and Grigg established a classification system of ORNTB, ranging from grade I (erosion of the EAC skin, without bony involvement) to IVb (skull base involvement) [8]. Finally, in 2014, Kammeijer, in his classification, divided localized disease and diffuse type A, with extensive necrosis of the temporal bone, with involvement of adjacent structures, but little pain and intact hearing, as well as diffuse type B, with similar features, but severe pain and infection and/or no functional hearing [9]. Our presented case certainly fell into a more severe category of diffuse disease with cranial nerve involvement.

There is currently a lack of randomized controlled trials regarding the condition, and, therefore, treatment needs to be personalized and aimed towards symptom control. The management options are wide ranging, including topical/systemic anti-microbiologic treatment, hyperbaric oxygen therapy, analgesia, debridement, and, in more severe cases, surgery in the form of lateral temporal bone resection, mastoidectomy, or subtotal petrosectomy, plus or minus flap obliteration [3,4]. In one study regarding parotid malignancies, interestingly, ORNTB after parotidectomy and RT was found in 1.9% of cases, but after parotidectomy, mastoidectomy and RT, it was found in 12.5%, and after combined parotidectomy, subtotal petrosectomy, and flap obliteration of the mastoid cavity and RT, it was found in 0% [10]. Generally, there seems to be consensus that in localized ORNTB with minor symptoms, nonsurgical (conservative) therapy is appropriate, but in patients with diffuse disease, cases refractory to conservative measures or major symptoms, surgery is more likely to be required [11]. The role of hyperbaric oxygen therapy in ORNTB is controversial. It may be considered as an adjunct, but a lack of evidence of its efficacy remains [11]. The outcomes of localized disease, which were treated conservatively, reported adequate resolution of symptoms in 89% of cases [4]. In the surgical group, the systematic literature review by Yuhan et al. described 93.8% of the subtotal of petrosectomies, 90.9% of lateral temporal bone resections, and 59.76% of mastoidectomies as successfully achieving the treatment goals [4]. Kammeijer et al. also proposed, in particular, subtotal petrosectomy, with a success rate of 90.9%, with vascularized flap obliteration [9].

The options for long-standing facial nerve rehabilitation are complex, including physical therapy and various surgical reanimation procedures [12]. These need to be scheduled in the context of the management of the underlying ORNTB and patients’ factors.

With evidence for treatments of the disease remaining currently limited, primary prevention should be taken into consideration. This would include particularly vascularised soft tissue reconstruction after ablative oncological procedures, avoiding tension over the temporal bone, and ensuring that no exposed bone is present within the radiation field at the beginning of radiotherapy. Finally, it has to be highlighted that in this case, RT was still planned with 3D conformal radiotherapy. New developments of intensity-modulated radiotherapy (IMRT), which is now established as a modern technique of radiation therapy, allow better control of radiation dose delivery in the head and neck. This has been shown to reduce, especially, radiation-induced xerostomia by sparing of salivary glands [13]. One hopes that ORNTB might be reduced as well in the future, through sparing of critical (exposed) bony structures.

Furthermore, the patient had a history of diabetes, with retinopathy and neuropathy. It is known that poor glycaemic control is associated with increased wound infections and postoperative morbidity, and, therefore, in head and neck cancer patients, optimization of diabetes is advised peri-operatively [14]. However, diabetic neuropathy, in contrast to an isolated facial nerve (motor) dysfunction, typically causes sensory dysfunction and pain, beginning distally in the lower extremities [15]. It has been reported that the incidence of cranial nerve palsies (including CN VII) is higher in diabetic patients, compared to non-diabetics; however, isolated cranial nerve VII palsies seem to be less closely related to diabetic complications than, for example, oculomotor neuropathies [16]. Therefore, diabetes, in the setting of clinically present ORNTB, might have been a contributary factor, but was, altogether, difficult to ascertain. Additionally, chemotherapy did employ fluorouracil, docetaxel and cisplatin in this case. Both docetaxel and cisplatin, and especially their combination, have been shown to frequently predominantly cause peripherally (lower and upper limbs) sensory neuropathies, but also motor neuropathies, usually within weeks of treatment [17,18]. Even though chemotherapy-induced neuropathies can progress after the completion of treatment, as the onset of the facial nerve dysfunction occurred more than seven years after chemotherapy, the influence of both drugs as causative agents appears unlikely in this case.

## 4. Conclusions

Osteoradionecrosis of the temporal bone is a delayed complication. It occurs several years after RT for head and neck cancer, and is a rare cause of facial nerve palsy. It is important to consider the diagnosis in patients with a history of mastoidectomy and RT affecting the ipsilateral temporal bone.

## Figures and Tables

**Figure 1 diagnostics-12-01021-f001:**
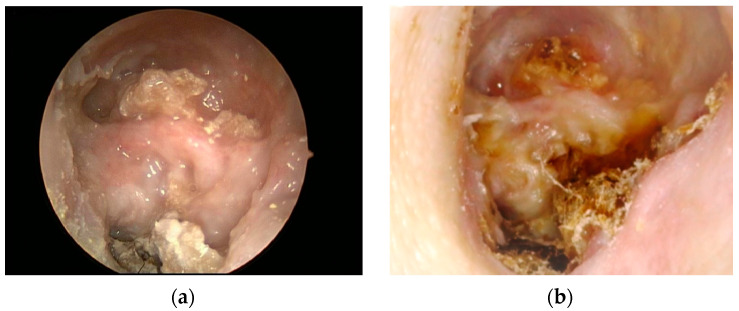
Left ear canal and mastoid cavity at initial presentation (**a**) and after six months, (**b**) with persistent, widely exposed necrotic bone on the floor of the external auditory meatus, facial ridge, and debris.

**Figure 2 diagnostics-12-01021-f002:**
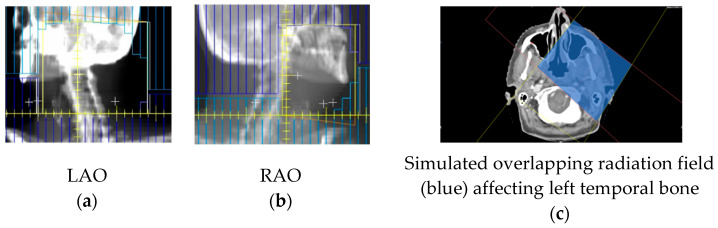
Upper radiation fields with 66 Gray (Gy) in 33 fractions to the left oropharynx tumor and neck. Radiotherapy was planned with 3D conformal radiotherapy, utilizing two fields to the upper volume: left anterior oblique (LAO) (**a**) and right anterior oblique (RAO) (**b**) fields, and the beam’s eye radiographs shown. These fields have been reconstructed on axial CT, to indicate the approximate volume (**c**). The area where the fields overlap (blue) received 66 Gy.

**Figure 3 diagnostics-12-01021-f003:**
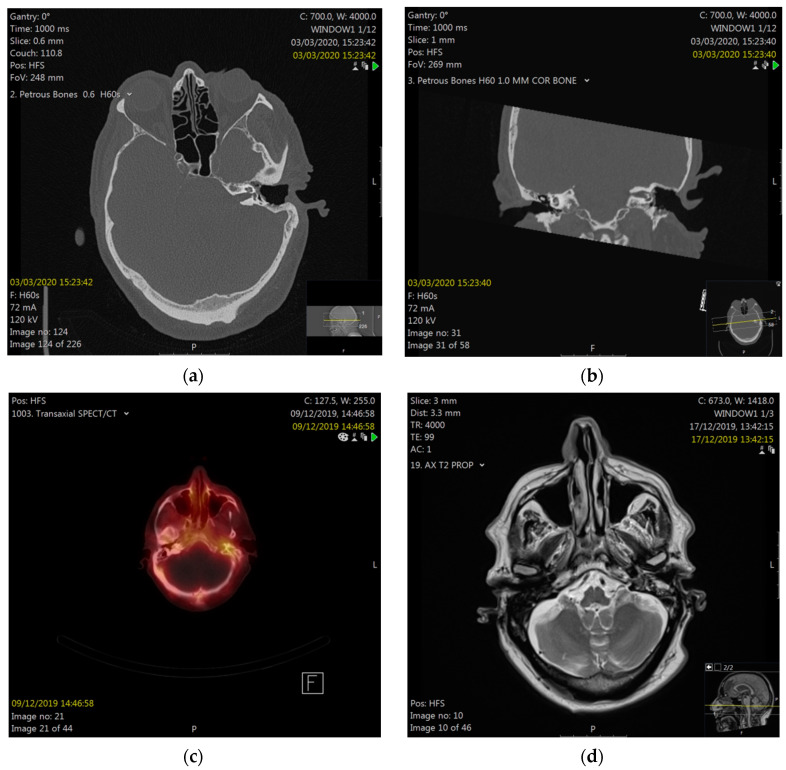
Computed tomography (CT) of petrous bones in axial (**a**) and coronal (**b**) planes, showing a situation after the left canal wall-down mastoidectomy, with patchy bony erosion of the floor of the external auditory meatus (white arrow) and uncovered facial nerve. Nuclear medicine scan (**c**) showing uptake around the left EAM. Magnetic resonance imaging (MRI) showing enhancement of the left mastoid (**d**).

## Data Availability

Not applicable.

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
