# Peer review of "Osteoradionecrosis of the Temporal Bone as a Rare Cause of Facial Nerve Palsy"

_diagnostics, 2022, doi:10.3390/diagnostics12051021_

Round 1

Reviewer 1 Report

The authors describe an interesting case-presentation where a patient with a history of a left CWD mastoidectomy underwent induction chemotherapy followed CRT for an oropharyngeal cancer and subsequently developed facial nerve dysfunction seven years later.

The authors postulate that the altered anatomy following the left CWD mastoidectomy predisposed the patient to radiation induced osteoradionecrosis and subsequently CN VII dysfunction. I feel the clinical history and work-up support the diagnosis of ORNTB and its plausible this directly or indirectly caused facial nerve failure.

I feel that the cases is informative and would be interest to head and neck oncologists. However, I feel the discussion regarding risk factors for this patient could be expanded. Therefore, I have several suggestions

  • In addition to the patients surgical history, the authors correctly identify that the use of 3D conformal radiotherapy may have contributed to the increased risk of osteoradionecrosis. However, the patient also received induction chemotherapy and concurrent chemoradiation. Please discuss the role these systemic agents may have had in this late effect of neuropathy.

  • Furthermore, please comment on the patients history of diabetes with retinopathy and neuropathy impact on this adverse effect.

Author Response

Dear colleague,

Thank you very much for your review and comments.

I have revised the manuscript including a discussion of the role of diabetes and chemotherapy regarding neuropathies. 

Please see the revised uploaded version.

Kind Regards,

Dr. Florian Schmidt

Reviewer 2 Report

The article is intended for a special issue - Evidence-Based Diagnosis and Management of Facial Nerve Disorders. The authors present an interesting case report of a patient with facial nerve impairment.

The case study is interesting and carefully processed, it is also very beneficial for practice, among other things from a perspective interdisciplinary point of view. Almost nothing can be said about the work, perhaps the authors should respect the recommendations of the journal for the structure of the article, which should be: Abstract, Keywords, Introduction, Detailed Case Description, Discussion, and Conclusions.

In connection with this, I recommend editing the individual titles of the work and add the Introduction part, which is missing.

I believe that despite the small scope of work, this case study is suitable for publication

Author Response

Dear colleague,

Thank you for your review and comments.

I have restructured the manuscript and added the introduction.

Kind Regards,

Dr Florian Schmidt